# Pharmacometabolomics Approach to Explore Pharmacokinetic Variation and Clinical Characteristics of a Single Dose of Desvenlafaxine in Healthy Volunteers

**DOI:** 10.3390/pharmaceutics16111385

**Published:** 2024-10-28

**Authors:** Anne Michelli Reis Silveira, Salvador Sánchez-Vinces, Alex Ap. Rosini Silva, Karen Sánchez-Luquez, Pedro Henrique Dias Garcia, Caroline de Moura Garcia, Rhubia Bethania Socorro Lemos de Brito, Ana Lais Vieira, Lucas Miguel de Carvalho, Marcia Ap. Antonio, Patrícia de Oliveira Carvalho

**Affiliations:** 1Health Sciences Postgraduate Program, São Francisco University–USF, Bragança Paulista 12916-900, SP, Brazil; anne.silveira@unifag.com.br (A.M.R.S.); salvador.vinces@mail.usf.edu.br (S.S.-V.); alex.rosini@mail.usf.edu.br (A.A.R.S.); ksanchezluquez@gmail.com (K.S.-L.); pedroh_g@outlook.com (P.H.D.G.);; 2Integrated Unit of Pharmacology and Gastroenterology (UNIFAG), São Francisco University–USF, Bragança Paulista 12916-900, SP, Brazil; marcia.antonio@unifag.com.br; 3Althaia S.A. Indústria Farmacêutica, Atibaia 12952-820, SP, Brazil; garcia.carol100@gmail.com (C.d.M.G.); rhubia.brito@althaia.com.br (R.B.S.L.d.B.);

**Keywords:** desvenlafaxine, metabolomics profile, metabolic pathways, pharmacokinetics

## Abstract

This study investigated the effects of a single dose of desvenlafaxine via oral administration on the pharmacokinetic parameters and clinical and laboratory characteristics in healthy volunteers using a pharmacometabolomics approach. In order to optimize desvenlafaxine’s therapeutic use and minimize potential adverse effects, this knowledge is essential. **Methods:** Thirty-five healthy volunteers were enrolled after a health trial and received a single dose of desvenlafaxine (Pristiq^®^, 100 mg). First, liquid chromatography coupled to tandem mass spectrometry was used to determine the main pharmacokinetic parameters. Next, ultra-performance liquid chromatography–quadrupole time-of-flight mass spectrometry was used to identify plasma metabolites with different relative abundances in the metabolome at pre-dose and when the desvenlafaxine peak plasma concentration was reached (pre-dose vs. post-dose). **Results:** Correlations were observed between metabolomic profiles, such as tyrosine, sphingosine 1-phosphate, and pharmacokinetic parameters, as well as acetoacetic acid and uridine diphosphate glucose associated with clinical characteristics. Our findings suggest that desvenlafaxine may have a broader effect than previously thought by acting on the proteins responsible for the transport of various molecules at the cellular level, such as the solute carrier SLC and adenosine triphosphate synthase binding cassette ABC transporters. Both of these molecules have been associated with PK parameters and adverse events in our study. **Conclusions:** This altered transporter activity may be related to the reported side effects of desvenlafaxine, such as changes in blood pressure and liver function. This finding may be part of the explanation as to why people respond differently to the drug.

## 1. Introduction

Depression is a common disorder, affecting 350 million people worldwide (WHO, www.who.int, accessed on 15 May 2023). Because this disorder is diverse and complex, its etiology and pathology remain unknown, as do the mechanisms of action of the antidepressants currently in use [1]. The exact nature of the adverse effects of antidepressant, their clinical significance, and potential risk factors still need to be fully elucidated. Of these, desvenlafaxine belongs to the serotonin–norepinephrine reuptake inhibitor (SNRI) class and is marketed as Pristiq^®^. It was approved by the FDA in 2008 to treat major depressive disorder (MDD) in adults. As a selective SNRI, desvenlafaxine does not significantly affect muscarinic–cholinergic, H1-histaminergic, or α1-adrenergic receptors, nor does it inhibit monoamine oxidase [2]. This selective inhibition is thought to contribute to its therapeutic effects in managing depression [3]. It clinical efficacy has been linked to increased levels of the neurotransmitters serotonin and norepinephrine in the central nervous system [4], with a good brain-to-plasma ratio, suggesting its utility in the central nervous system, peripheral nervous system, and peripheral-related disorders associated with changes in these neurotransmitters. However, there has been no research into its effects on individuals at a more systematic level.

Structurally, desvenlafaxine succinate (DVS), an active metabolite of venlafaxine, is commercialized as the succinate salt monohydrate of O-desmethylvenlafaxine (MW: 263.1885 g/mol, C_16_H_25_NO_2_) that can be excreted unchanged (~45%), as the glucuronide metabolite (~19%), and <5% as the oxidative metabolite (N,O-didesmethylvenlafaxine). It is oxidized primarily by CYP3A4 [5]. Upon oral intake, its bioavailability stands at 80%, reaching maximum plasma concentrations (C_max_) after 7.5 h, and it shows low plasma protein binding (30%). The distribution of the drug after intravenous administration suggests extensive non-vascular compartmentalization, with a volume of distribution of 3.4 L/kg and a mean half-life of approximately 11 h [5]. Traditional pharmacokinetic prediction methods [6] are not always able to predict the behavior of the drug in a given individual, and inter-individual variability in response to this drug remains a clinical challenge [7]. Pharmacokinetic and pharmacometabolic studies are essential to improving our understanding of the mechanisms underlying this variability. The current literature, although comprehensive, could benefit from further exploration of metabolomics. Metabolites, which are the chemical entities that are transformed during metabolism, provide a functional readout of the cellular biochemistry that is the best predictor of the phenotype [5]. It justifies the relevance of a pharmacometabolic approach to explore the pharmacokinetic variability of desvenlafaxine in healthy volunteers.

Understanding the additional pathways that may be affected in addition to the direct mechanism of action of DVS is important for effective clinical management. Inter-individual variability response to desvenlafaxine, elucidated by a metabolomics approach, is extremely useful in obtaining a picture of the physiological state of a given individual [8]. The present study examines the association between variation in the untargeted metabolomic profile and pharmacokinetic and clinical parameters of a unique dose of DVS in healthy volunteers. Community network modeling and enrichment analysis were employed and integrated to interpret how sets of metabolites and their functions can explain interindividual variability. These findings could help to understand how DVS can be used to maximize effectiveness and minimize side effects in order to develop personalized strategies.

## 2. Materials and Methods

### 2.1. Healthy Human Volunteers and Study Design

The present study was nested in a single-dose, open-label, randomized, pharmaco-bioequivalence study performed under biosafety and ethical standards. The data used were derived from 35 healthy volunteers who received the reference drug (Pristiq^®^, by Pfizer, Newbridge–Ireland) and who were selected for the study after assessment of their health status via clinical evaluation and a comprehensive set of laboratory tests. The same parameters were used to monitor adverse events post study.

### 2.2. Ethics

The protocol complied with the current Brazilian legislation on clinical research in humans and was duly approved by the Research Ethics Committee. Institutional Review Board (IRB) approval for the study was received under protocol number 58633322.3.0000.5514, dated 1 September 2022. All subjects gave their written informed consent and were free to withdraw from the study at any time.

### 2.3. Determination of Pharmacokinetic Profile

The analytical method was validated according to the respective resolution of the Agência Nacional de Vigilância Sanitária [Brazilian Public Health Surveillance Agency] [9]. The study used an LC-MS/MS system to verify quantification and calculate pharmacokinetic parameters and performed chromatographic separations in an LC-20AD analytical pump (Shimadzu-Kyoto, Japan) using a SIL-20A HT autosampler (Shimadzu-Kyoto, Japan), coupled with mass spectrometry analyses using a Quattro Micro (Micromass, Newcastle, UK) mass spectrometer equipped with an electrospray ionization (ESI) source. Desvalexine was detected via a multiple reaction monitoring (MRM) transition of 264.33 > 58.15, and orphenadrine was detected via an MRM transition of 270.32 > 181.26. Data was acquired using MassLynx 4.1 (Waters, Newcastle, UK). More information on the validation method can be found in the Appendix A, including chromatograms for the selectivity assay (Appendix A). Pharmacokinetic parameters were calculated employing a non-compartmental statistic using WinNonLin 8.3 software (Pharsight, New Jersey, USA).

### 2.4. Metabolomic Profile

The samples were collected at two times: (i) baseline samples (0 h), named the pre-dose group; and (ii) samples at Tmax, named the post-dose group. The samples were randomized, and 25 µL of each sample was collected to compose a pooled sample used as the Quality Control (QC). An aliquot of plasma (200 μL) was extracted by adding cold methanol (400 μL) for protein precipitation. Samples were then shaken for 30 s for protein precipitation and centrifuged at 12,000 rpm for 10 min at 4 °C. Finally, the supernatant (400 μL) was collected and dried over a nitrogen gas (N_2_) flow. Samples were resuspended in 200 µL of acetonitrile (ACN)/water (H_2_O) (1:1 *v*/*v*). Sample extraction and analysis were conducted randomly to reduce instrumental and biological bias. The QCs were analyzed after each 10 samples to check extraction and system stability deviations.

The analysis was performed in an ACQUITY^®^ FTN H Class (Waters, Newcastle, UK) liquid chromatography (UPLC) coupled to a XEVO-G2XS (QToF) quadruple time-of-flight mass spectrometer (UPLC-QToF-MS^E^) (Waters, Newcastle, UK) equipped with an electrospray ionization (ESI) source operated in the positive and negative ionization mode, separately. An ACQUITY UPLC ^®^ CSH C18 column (2.1 mm × 100 mm × 1.7 µm, Waters, Newcastle, UK) was employed as a stationary phase [10]. More information about the metabolomic analysis can be found in the Appendix A and Methods.

#### 2.4.1. Data Processing and Feature Selection

Processing of the LC-MS raw files used the Progenesis™ QI software v2.4 (Non-linear Dynamics, Newcastle, UK), enabling adduct selection, peak alignment, deconvolution, and compound annotation via MS^E^ experiments. Progenesis QI generates an intensity table of the features, labeled according to their nominal masses and retention times as a function of their intensity. The SERRF package [11], a QC-based signal correction method implemented in R 4.2.3 statistical programming language [12], was used to correct any potential signal drift. Then, zero values were replaced by half of the smallest positive value in the original dataset. Features with constant or near-constant values were filtered out using an interquartile range (lowest 10% by IQR rank) criterion. Features with relative standard deviation (RSD) in QC sample data over 20% were filtered out. Filtered data were transformed using a logarithmic scale and a Pareto scale.

To select the features with differential abundance between pre-dose and post-dose groups, the researchers used the limma method [13] implemented in the MetaboAnalystR 3.0 [14] package in R. First, the covariates or confounding variables were selected according to the literature [15] for known characteristics: sex, age, and body mass index (BMI). The period variable from the bioequivalence study was considered a covariate too. The participant tag variable was considered to be a blocking factor in order to preserve the relationship between samples before and after desvenlafaxine administration.

#### 2.4.2. Metabolite Identification

Due to low- and high-energy acquisition enabled by the use of MS^E^, we have information on precursor ions (low energy) and fragments (high energy) in the same spectrum. Annotation of molecules considered mass accuracy (≤5 ppm), fragmentation profile (≤10 ppm), isotope similarity, and biological relevance. The in-house “SDF2PQI” software v1.0 was employed to enhance fragment match numbers for Progenesis PQI data compatibility with external SDF-based spectral libraries [16]. The libraries used were MassBank of North America (MoNA) [17], the Human Metabolome Database (HMDB) [17], and the LIPID MAPS structure database [18].

#### 2.4.3. Pathway Enrichment

Knowledge-driven pathway enrichment was the first approach to interpreting the differential set of metabolites. It used the RaMP 2.0 package [19], developed in R. This tool queries 3 different databases for set enrichment: KEGG [20], WikiPathways [21], and Reactome [22]. The RaMP parameters used were 2 minimum hits per pathway, pathways with at least 5 and up to 150 metabolites, and the FDR value ≤ 0.05. The results contain a list of enriched pathways, source and ID, statistical values, and matched and complete metabolite lists.

To facilitate the understanding of the list of enriched pathways, we considered a grouped top-level and subpathways hierarchy as described by the Reactome database. That led us to hypothesize that the pathway hierarchy with most members could be helpful in acquiring a better understanding of the enriched biological or structural function [23].

#### 2.4.4. Community Network Analysis

Metabolic processes imply that there is interdependence of the abundances of metabolites, evidenced by their correlated values. That association can be extended to causal, biological, or chemical relationships [24]. Here, we calculated the degree of association between individual metabolites and each PK parameter or patient characteristic using the distance correlation metric [25]. The distance correlation calculates the linear correlation between variables, like the Pearson correlation, but also a distance-based non-linear correlation with values from 0 to 1. Additionally, the Pearson correlation was calculated to obtain a sign or direction for each pair’s distance correlation. Only correlations (d) ≥ 0.5 and with adjusted FDR *p*-value ≤ 0.05 were considered for further analysis.

A similarity matrix based on those correlations was used to construct a tripartite network, where metabolites are the connectors between PK parameters and patient characteristics. The correlation values were used as weighted edges, and the metabolites, PK parameters, and patient characteristics were used as nodes in the network. The Leiden algorithm was used to find community structures in our graph. The igraph package [26] was used to determine the graph structure and the modules or communities.

## 3. Results

### 3.1. Clinical and Laboratory Characteristics of Volunteers

At the beginning and end of the study, all participants underwent clinical evaluation and laboratory tests using the reference values proposed by ANVISA or without clinical significance, according to a medical report. A complete summary of participants’ characteristics is available in Appendix A. Clinical and laboratory characteristics outside the normal referential criteria were reported as adverse events (Appendix A).

### 3.2. Determination of Desvenlafaxine Pharmacokinetics

Table 1 presents a summary of descriptive statistics of the pharmacokinetic parameters, area under the curve (AUC), maximum concentration (C_max_), time to maximum concentration (T_max_) (h), elimination rate constant (Kel) (1/h), and half-life (T ½)( h). Figure 1 shows the individual pharmacokinetic curve (concentration vs. time). Appendix A shows the complete data for concentration versus time and PK parameters.

### 3.3. Metabolomic Analysis

The untargeted analysis resulted in a total of 2671 features detected in positive mode and 4634 in negative mode. Our feature selection model found 1548 features in negative ion mode and 623 features in positive ion mode as differentials. From those 2171 differential features, 238 were putatively identified as metabolites. Figure 2A,B represents the dispersion of participants and QC samples using principal components. Figure 2C,D shows the heatmaps for the relative abundance of features selected and identified. Compound and identification data obtained for the selected and identified metabolite are reported in Appendix A.

### 3.4. Enrichment Analysis

The 238 selected and identified features were used as input for the pathway enrichment analysis. There were 36 pathways with an adjusted *p*-value by FDR ≤ 0.05. Appendix A contains the complete results as generated by the RaMP package. The highly related pathways in the results are from the Reactome database. Some Wikipath pathways are their own versions of the Reactome pathways, but some original ones are related to those with the lowest FDR. Due to the high number of shared metabolite members, we used a top-level and sub-pathway hierarchy suggested by the Reactome tool as additional information to interpret the results. That makes it possible to understand and discuss the main functional or structural characteristics between enriched pathways.

Top-level hierarchies point to an enriched set of pathways related to the activity and disturbance of SLC and ABC proteins and their function as transporters of small molecules, especially those related to bile acids, nucleotides, vitamins, and related structures.

### 3.5. Community Network Analysis Results

After module detection using community network analysis, 31 modules were evidenced, representing sets of metabolites associated with PK parameters and participant characteristics. In the pre-dose metabolome dataset, 57 metabolites were found to be significantly associated with 4 PK parameters, 3 clinical characteristics, 22 characteristics in serum, and 2 in urine related by 95 edges that formed 17 modules. In the post-dose metabolome dataset, 66 metabolites were found to be significantly associated with 5 PK parameters, 3 clinical characteristics, 17 characteristics in serum, and 1 in urine related by 121 edges that formed 16 modules.

Figure 3A represents the community networks in the pre-dose group and Figure 3B represents the community network in the post-dose group. The complete lists of members for communities at the pre-dose and post-dose stages can be found in Appendix A, respectively.

### 3.6. Integrating Pathway and Network Results

These joint results showed that PK parameters were associated with metabolites involved in 3 top-level pathways related to the transportation of small molecules and signal transduction; 12 subpathways related to the 3 top-level pathways, especially to SLC and ABC transporters and the ones related to signaling via GPCR, nuclear receptors, and receptor tyrosine kinase; and to the metabolism of small molecules. The matches between the members of the metabolite sets of the modules and the members of the metabolite sets obtained in the enrichment with the individual characteristics association are listed in Table 2.

Results are summarized according to metabolite. Second columns indicate associated pharmacometrics and participant characteristics, either at baseline or after desvenlafaxine administration; last columns indicate top and sub pathway defined by enrichment analysis. Abbreviations: area under the curve (AUC), maximum concentration (C_max_), time to maximum concentration (T_max_) (h), elimination rate constant (Kel) (1/h) and half-life (T1/2) (h), Identificator Human Metabolome Database (HMDB), Reactome Homo Sapiens pathway (R-HSA), WikiPathways (WP), (RDW) variation in the size of red blood cells (RBCs), Aspartato aminotransferase (AST).

## 4. Discussion

In this study, we used a knowledge-driven approach based on metabolite set enrichment and a data-driven approach based on community network analysis to evidence and understand the relationship between the pharmacokinetic parameters and clinical and laboratory characteristics of participants and the differential metabolomic profile of healthy participant plasma samples before and after the exposure to a single dose of desvenlafaxine. The interpretation of our results must be defined as a significant association with the metabolome profile before, after, or in both situations of exposure to the drug.

No significant differences were observed in the participants’ baseline demographic parameters. Some common and mild adverse effects were reported, as well as more uncommon effects, as discussed below. The pharmacokinetic parameters obtained in this study are comparable to those previously reported for desvenlafaxine in the literature [27].

### 4.1. Association to Pharmacokinetics Parameters

Desvenlafaxine is a well-known selective serotonin–norepinephrine reuptake inhibitor that targets the transporters SLC6A2, SLC6A3, and SLC6A4. However, solute carrier (SLC) transporters, a class of proteins responsible for transporting ions, nutrients, and drugs, remain relatively underexplored. While there is no homology between different SLC families, members within the same family share 20–25% of their amino acid sequences [28].

In the pre-dose moment, AUC and C_max_ were correlated by the putatively annotated secondary metabolite N-(1-deoxy-1-fructosyl)phenylalanine and hydroperoxylinoleic acid. This phenylalanine derivative was negatively correlated with the AUC parameter but is poorly described in the literature. Hydroperoxylinoleic acid is part of a set of metabolites that enrich linoleic acid oxylipin metabolism as a product of the direct metabolism of n-6 polyunsaturated fatty acids (PUFA) [29], and its abundance was positively correlated with these PK parameters.

The post-dose evaluation shows that the abundance of tyrosine was positively correlated with AUC and C_max_ as a possible effect on desvenlafaxine abundance. Here, the tyrosine participated in a set of metabolites that enriched pathways related to the human solute carrier (SLC) transporters. Some SLC transporters, such as SLC6A194, SLC16A105, and SLC7A86, are plasma membrane proteins that facilitate the uptake of amino acids such as L-tyrosine and are particularly abundant in the kidney. Some studies indicate the importance of tyrosine kinase in the expression, function, and stability of SLC proteins [30,31], and the increased abundance of tyrosine is probably due to inhibition of their SLC transporters. In addition, sphingosine-1-phosphate is enriched in some interesting pathways that phosphorylate sphingosine, such as vascular endothelial growth factor receptor 2 (VEGFR2)-mediated cell proliferation and extra-nuclear estrogen signaling, which could explain the increased cell recruitment [32] in the blood of participants with relatively higher AUC or C_max_, considering the positive correlation between S1P and these parameters in post-dose vs. pre-dose moments.

Looking at the set of metabolites associated with this T_max_ in the post-dose moment, we found the o-desmethylvenlafaxine glucuronide, a glucuronide conjugate product, was positively correlated, so a delayed time to reach the maximum concentration is related to a higher abundance of this metabolite. Other members of this set are from different lipid classes, like palmitoleic acid, together with other relevant metabolites from this set, that enriched class A/1 (rhodopsin-like receptors) by interacting with the G protein-coupled receptor 120 (GPR120) [33] and the G alpha (q) signaling event as part of the ligands that activate the G alpha q protein (or Gq/11) [34]. However, we found no direct evidence related to desvenlafaxine in the literature.

The elimination rate constant (Kel) and half-life before and after drug exposure were positively correlated with tryptophol. Tryptophol is a tryptophan catabolite converted by the gut microbiota and absorbed by the intestinal epithelium [35], which could mean that the observed difference in relative abundance is not directly affected by desvenlafaxine, but its presence positively conditions or reflects the participants’ ability to eliminate it in both pre- and post-dose situations.

Although our results demonstrated sets of metabolites associated with some PK parameters, further studies are needed to determine how this may condition the achieved AUC or C_max_ for each participant and how differences in desvenlafaxine blood concentration could affect some cellular functions.

### 4.2. Association with Clinical and Laboratory Characteristics

In previous clinical trials, desvenlafaxine was associated with several mild adverse effects, like nausea, as well as less common, but more serious, adverse effects like some which are cardiovascular related [36]. Increased blood pressure was reported in five participants (10%) of a clinical study [37]. In another study, short-term desvenlafaxine treatment was associated with small but statistically significant increases in systolic blood pressure and diastolic blood pressure [38], evidencing the need for more research on the safety and efficacy of desvenlafaxine in specific patient populations such as adults with cardiovascular disease [37]. Desvenlafaxine may cause adverse effects in patients with kidney or liver problems and can increase blood cholesterol and triglyceride levels [39]. These previous clinical reports are consistent with the results of our community network analysis, which suggest associations between pre- and post-dose variability in metabolite abundance and clinical parameters related to blood pressure, heart rate, serum lipids, and markers of liver activity, and are evident in pathways highly enriched for all differentially selected metabolites.

Transport of small molecules mediated by SLC transmembrane transport and ABC-family proteins are of the most represented pathways by the selected metabolites sets, related to the solute carrier (SLC) [40] superfamily of transporters and the ATP-binding cassette (ABC) [41] transporters.

As described in associations with PK parameters, our results show that SLC transporter activity is altered by the presence of desvenlafaxine. It is challenging to specify a pattern among the transporter proteins involved in the pathways enriched by the selected differential metabolite sets given that they are involved in similar biological functions. To date, eight SLC gene families (SLC1, SLC6, SLC7, SLC16, SLC25, SLC36, SLC38, and SLC43) have been found to be involved in the transportation of amino acids and oligopeptides. In addition, another group (SLC13, SLC22) can transport bile salts, organic acids, metal ions, and amine compounds. SLC superfamilies are widely expressed throughout the body, most notably in the epithelia of major organs, such as the liver, intestine, and kidney [28]. On the other hand, ABC proteins, another family of transporters, harness energy from ATP hydrolysis and function as efflux transporters [41]. Most of the differential metabolites are substrates for the SLC protein transporters. For the transport of vitamins, nucleosides, and related molecules pathway, 13 metabolites of a total of 62 were present in our results, and 5 were associated with one or more phenotypes, as inferred from the community network analysis.

Oleic acid, an unsaturated C-18 fatty acid, in pre-dose, is a member of a network-derived module associated with the number of leukocytes in blood and urine, the number of segmented neutrophils, and the levels of aspartate aminotransferase in blood. This lipid is a substrate of the SLC27A1,4,6 that transports LCFAs from the extracellular region to cytosol. These fatty acid transporter proteins (FATP) have been shown to transport the prototypical LCFA oleic acid (OLEA), and the FATP1 is highly expressed in adipose tissue and muscle [42]. FATP4 is the major intestinal LCFA transporter [43] but is also expressed at lower levels in the brain, kidney, liver, and heart. FATP6 is localized to cardiac myocytes [44]. Linoleic acid is also transported by the SLC27 protein and is associated with RDW at the pre-dose and at post-dose moments, as well as with hemoglobin, hematocrit, erythrocytes count, and systolic pressure as a cardiovascular-related phenotype in addition to serum creatinine and albumin.

At pre-dose, another metabolite set with similar phenotype associations as the previous one is related to renal activity (serum creatinine and uric acid), blood cell recruitment (serum hemoglobin, hematocrit, erythrocytes, and eosinophils cell number), and hepatic activity (serum alkaline phosphatase). A member of this module, uridine diphosphate glucose (UDP-D-glucose), is a hub metabolite at a network level and appears related to the pathways of the three main classes of receptors: SLC, ABC, and rhodopsin-like receptors. At post-dose, uridine diphosphate glucose is a member of a similar module without the association with white blood cell count but with an additional association with the participant’s systolic pressure. Uridine diphosphate glucose is a pyrimidine nucleotide sugar that acts as an intermediate in carbohydrate metabolism transported by the SLC35D2 that exchanges UDP-D-glucose for UMP and resides on the Golgi membrane where it mediates the transportation of nucleotide sugars [45]. Recent evidence suggests that increased plasma levels of UDP-D-glucose could trigger an inflammatory-like response in some tissues and blood cells [46,47]. UDP-D-glucose, uric acid, and isolucylproline are metabolites present in the two modules previously commented upon, with similar associations at the pre-dose and post-dose moments, evidencing that desvenlafaxine activity increased their plasma levels relative to their initial levels before drug administration. Androsterone sulfate, an androsterone metabolite, is a member of the same module as UDP-D-glucose at pre-dose and is positively correlated with serum uric acid before and after drug exposure.

Another SLC group involved in the transport of organic anions, the SLCO1A2, mediates the transportation (influx) of bile salts in the plasma membrane and may play a minor role in the uptake of bile salts and acids by the liver [48]. Bile salts and acids exist in the blood as complexes with serum albumin, and their uptake by SLCO1A2 must involve the disruption of that complex. Chenodeoxyglycocholic acid, a glycine that conjugates bile acid, is negatively correlated to the heart rate of the participants.

Among the pathways related to ABC protein transporters, the defective ABCB11 pathway causes benign recurrent intrahepatic cholestasis type 2 (BRIC2) was highly enriched in our results. This mediates the release of bile salts from liver cells into bile [48]. In addition to the bile salt (chenoglycodeoxycholic acid glycine conjugate, chenoglycodeoxycholic acid, glycocholic acid, and taurocholic acid), our analysis found the adenosine triphosphate to be a differential metabolite that enriched this pathway. As mentioned before, chenoglycodeoxycholic acid is associated with the heart rate of participants post-drug exposure.

### 4.3. Signal Transduction: Class A/1 (Rhodopsin-like Receptors)

The Rhodopsin-like receptors (class A/1) pathway was enriched by 12 selected metabolites, of which 3 were associated with some participant characteristic values, as inferred from the community network analysis. All the associations were relevant to the metabolomes at pre-dose, which can be understood as a basal metabolomics profile that conditioned the phenotype status but lost its quantifiable correlation due to desvenlafaxine in the blood. Associated with white blood cell counts and serum aspartate aminotransferase, oleic acid is an extracellular ligand for the free fatty acid receptors 1 and 4 (FFAR1-4) [49]. Another lipid, lysophosphatidic acid, is a glycerophospholipid ligand to G alpha q protein (Gq/11) and is associated with serum total cholesterol levels. The activation of this protein leads to responses that are still not well understood [50]. Sphingosine-1-phosphate binds to receptor de esfingosina-1-fosfato 1 and 2 (S1PR1-2). The S1PR2 participates in S1P-induced cell proliferation, survival, and transcriptional activation, effects mediated by coupling to Gi and Gq proteins; while S1PR1 has an important role in regulating endothelial cell cytoskeletal structure, migration, capillary-like network formation, and vascular maturation [51]. This signaling activity could explain the phenotype variation that is also related to the desvenlafaxine C_max_, as commented on in the PK parameter discussion. Network analysis also reported other characteristics associated with metabolites for which we found no evidence of their association in the current literature. Further research is crucial to explore these connections and develop strategies by which to optimize treatment efficacy while minimizing adverse effects.

This study sheds light on the complex interplay between desvenlafaxine and the human body’s metabolic landscape. The presence of desvenlafaxine appears to have a broader selectivity than previously reported, affecting a diverse group of SLC transporters as well as ABC transporters. By extension, using network analysis, pathways involving SLC transporters could also be related to variations associated with the adverse effects of this drug. This could be used to understand the inter-individual variability observed in populations using this drug. Specific studies are needed to validate this information and to consider genetic aspects that may influence these pharmacophenotypes.

## Figures and Tables

**Figure 1 pharmaceutics-16-01385-f001:**
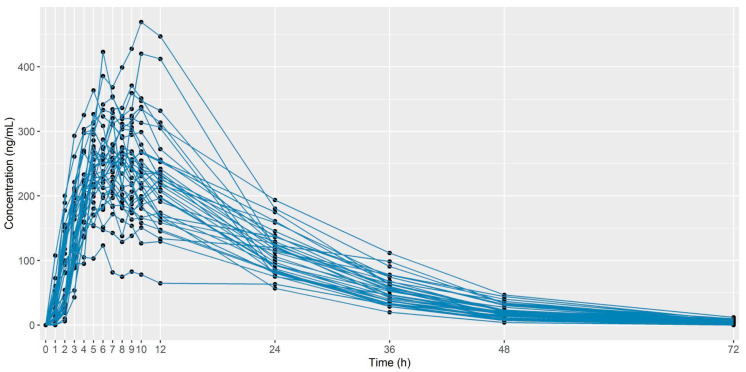
Plots individual pharmacokinetics curve (concentration vs. time).

**Figure 2 pharmaceutics-16-01385-f002:**
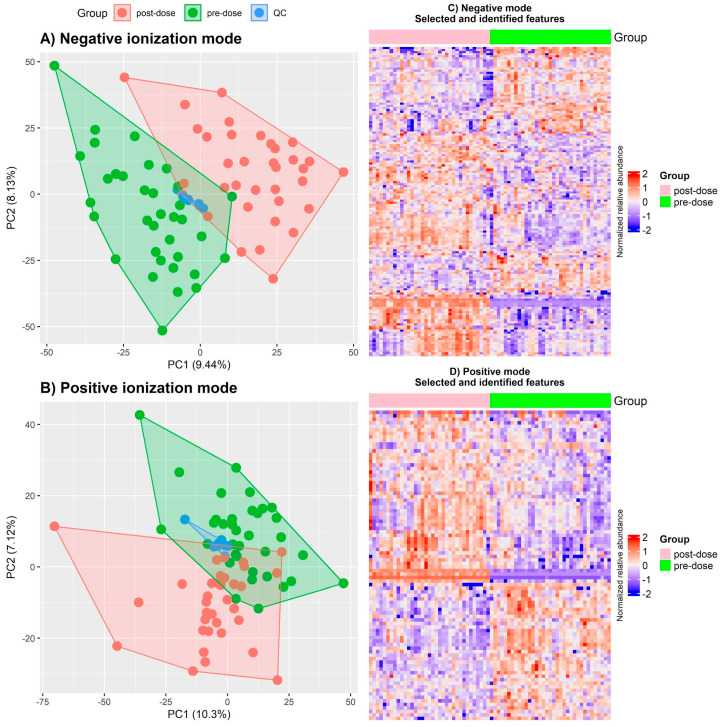
PCA plots for (**A**) negative and (**B**) positive ionization modes and heatmaps of relative abundance for selected and identified features of (**C**) negative and (**D**) positive ionization modes.

**Figure 3 pharmaceutics-16-01385-f003:**
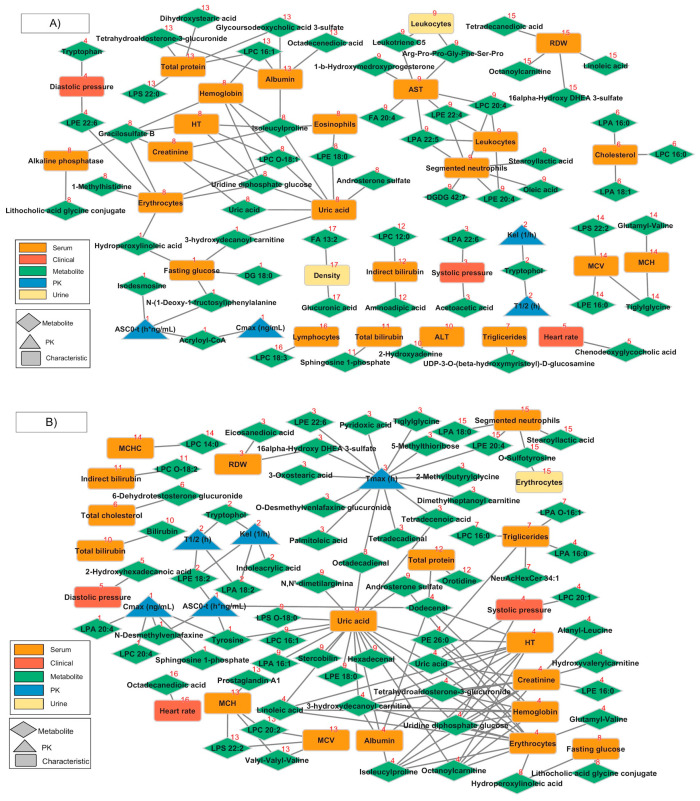
Community networks were obtained using PK parameters, clinical and laboratory characteristics, and identified metabolites at (**A**) pre-dose and (**B**) post-dose. Communities or modules are numbered with a red label above each node.

**Table 1 pharmaceutics-16-01385-t001:** Descriptive statistics of the pharmacokinetics parameters. Descriptive statistical values (in columns) for the five pharmacokinetics parameters (in rows) calculated in the study.

PK Parameters	Mean	Minimum	Median	Maximum	Standard Deviation	Coefficient of Variation (%)
**C_max_ (ng/mL)**	290.29	123.16	275.90	469.15	74.81	25.78
**AUC 0-t (h*ng/mL)**	6051.15	2711.68	5783.32	10,338.49	1522.49	25.16
**T ½ (h)**	9.43	6.63	9.35	12.00	1.32	14.02
**T_max_ (h)**	7.40	4.00	7.50	12.00	1.82	24.60
**Kel (1/h)**	0.08	0.06	0.07	0.11	0.01	14.76

Legend: area under the curve (AUC); maximum concentration (C_max_); time to maximum concentration (T_max_) (h); elimination rate constant (Kel) (1/h); and half-life T ½( h).

**Table 2 pharmaceutics-16-01385-t002:** Integrated pathways and community network analysis results.

Parameters/Characteristics	Group	Metabolite Hits	Top-Level Pathway (Subpathway)
Pharmacokinetic Parameters			
AUC 0-t (h*ng/mL) C_max_ (ng/mL)	Post-dose	Tyrosine	Disease (disorders of transmembrane transporters)/Metabolism of proteins (Translation)/Transport of small molecules (SLC-mediated transmembrane transport)
		Sphingosine 1-phosphate	Signal transduction (signaling by GPCR/Signaling by nuclear receptors/Signaling by receptor tyrosine kinases)
Kel (1/h) T ½( h)	Pre-dose	Tryptophol	Individual evidence in the literature (N/A)
	Post-dose		
T_max_ (h)	Post-dose	O-Desmethylvenlafaxine glucuronide	Individual evidence in the literature (N/A)
		Palmitoleic acid	
			Metabolism of proteins (peptide hormone metabolism)
**Clinical characteristics**			
Systolic pressure	Pre-dose	Acetoacetic acid	Disease (disorders of transmembrane transporters)/Transport of small molecules (SLC-mediated transmembrane transport)
	Post-dose	Uridine diphosphate glucose	Disease (disorders of transmembrane transporters)/Signal transduction (signaling by GPCR)/Transport of small molecules (SLC-mediated transmembrane transport)
		Uric acid Linoleic acid	Disease (disorders of transmembrane transporters)/Transport of small molecules (SLC-mediated transmembrane transport)
Diastolic pressure	Pre-dose	Tryptophan	Disease (disorders of transmembrane transporters)/Metabolism of proteins (Translation)/Transport of small molecules (SLC-mediated transmembrane transport)
Heart Rate	Pre-dose	Chenodeoxyglycocholic acid	Disease (disorders of transmembrane transporters)/Transport of small molecules (SLC-mediated transmembrane transport)/Metabolism (metabolism of lipids)/Digestion and absorption (Digestion)
**Laboratory characteristics**			
Serum total cholesterol	Pre-dose	LPA 18:1	Signal transduction (Signaling by GPCR)
	Post-dose	6-Dehydrotestosterone glucuronide	Metabolism (metabolism of nucleotides)
AST Leukocytes Segmented neutrophils Leukocytes in Urine	Pre-dose	Oleic acid	Disease (disorders of transmembrane transporters)/Signal transduction (signaling by GPCR)/Transport of small molecules (SLC-mediated transmembrane transport)/Metabolism of proteins (peptide hormone metabolism)
Total bilirubin	Pre-dose	Sphingosine 1-phosphate	Signal transduction (signaling by GPCR/Signaling by nuclear receptors/Signaling by receptor tyrosine kinases)
	Post-dose	Bilirubin	Disease (disorders of transmembrane transporters)/Metabolism (metabolism of porphyrins)
Indirect bilirubin	Pre-dose	Aminoadipic acid	Cerebral organic acidurias, including diseases (N/A)
Serum Creatinine Uric acid Alkaline phosphatase Serum Hemoglobin Serum Erythrocytes Serum Hematocrit Serum Eosinophils	Pre-dose	Uridine diphosphate glucose	Disease (disorders of transmembrane transporters)/Signal transduction (signaling by GPCR)/Transport of small molecules (SLC-mediated transmembrane transport)
		Uric acid	Disease (disorders of transmembrane transporters)/Transport of small molecules (SLC-mediated transmembrane transport)
		Androsterone sulfate	Signal transduction (signaling by GPCR/Signaling by nuclear receptor/by Receptor tyrosine kinases)/Transport of small molecules (SLC-mediated transmembrane transport)/Metabolism (metabolism of porphyrins)
RDW	Pre-dose	Linoleic acid	Disease (disorders of transmembrane transporters)/Transport of small molecules (SLC-mediated transmembrane transport)
Fasting glucose	Post-dose	Lithocholic acid glycine conjugate	Metabolism (lipid metabolism)
		Hydroperoxylinoleic acid	Linoleic acid oxylipin metabolism (N/A)
Serum uric acid	Post-dose	Stercobilin	Metabolism (Metabolism of porphyrins/Metabolism of lipids)
		Hexadecenal	
		N,N’-dimetilarginina	Signal transduction (signaling by nuclear receptors/Signaling by receptor tyrosine kinases)

## Data Availability

The data presented in this study are available on request from the corresponding author.

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
