# Peer review of "Pharmacometabolomics Approach to Explore Pharmacokinetic Variation and Clinical Characteristics of a Single Dose of Desvenlafaxine in Healthy Volunteers"

_pharmaceutics, 2024, doi:10.3390/pharmaceutics16111385_

Round 1
Reviewer 1 Report
Comments and Suggestions for Authors
In this paper, the authors present the use of pharmacometabolomics approach to understand the pk variation and clinical characteristics following a single dose of desvenlafaine in healthy volunteer. Overall, this article is pretty informative. Looking at the metabolites and their functions to understand interindividual variability is quite useful. The authors have provided a very comprehensive introduction. However, it is quite difficult for me to evaluate the results since all figures mentioned in the article are missing. I would appreciate if the authors could look into this issue and would love to give more comments or suggestions once the missed figures are provided. Please see some minor comments I have as below for right now:
1) Line 29, please check the grammar for "Next, was used ultra-performance...."
2) In Table S1, the row of sex has mean 35 and standard deviation 51-49. Could you please give more clarifications around that?
3) From Line 303 to line 342, the author discussed about the association to PK. But no data in the article have indicated any correlation. I would appreciate more clarification.
Comments on the Quality of English LanguageIn this paper, the authors present the use of pharmacometabolomics approach to understand the pk variation and clinical characteristics following a single dose of desvenlafaine in healthy volunteer. Overall, this article is pretty informative. Looking at the metabolites and their functions to understand interindividual variability is quite useful. The authors have provided a very comprehensive introduction. However, it is quite difficult for me to evaluate the results since all figures mentioned in the article are missing. I would appreciate if the authors could look into this issue and would love to give more comments or suggestions once the missed figures are provided. Please see some minor comments I have as below for right now:
1) Line 29, please check the grammar for "Next, was used ultra-performance...."
2) In Table S1, the row of sex has mean 35 and standard deviation 51-49. Could you please give more clarifications around that?
3) From Line 303 to line 342, the author discussed about the association to PK. But no data in the article have indicated any correlation. I would appreciate more clarification.
Author Response
Thank you very much for taking the time to review our manuscript. We appreciate all the suggestions you made. Please find our detailed responses below, with the corresponding revisions and corrections highlighted in red in the resubmitted manuscript. We would like to apologize for the problems with visualizing the figures in the manuscript. In fact, we forward them individually and separately to the text, and they end up not being viewed. We regret any confusion this may have caused.
Point-by-point response
Comments 1: In this paper, the authors present the use of pharmacometabolomics approach to understand the pk variation and clinical characteristics following a single dose of desvenlafaxine in healthy volunteers. Overall, this article is pretty informative. Looking at the metabolites and their functions to understand interindividual variability is quite useful. The authors have provided a very comprehensive introduction. However, it is quite difficult for me to evaluate the results since all figures mentioned in the article are missing. I would appreciate it if the authors could look into this issue and would love to give more comments or suggestions once the missed figures are provided.
Response 1: We thank the reviewer for their positive feedback.
Comments 2: Line 29, please check the grammar for "Next, was used ultra-performance...."
Response 2: We appreciate the reviewer's comments. As suggested, the grammar error was corrected.
Old Sentence: Next, was used ultra-performance liquid chromatography-quadrupole time-of-flight mass spectrometry to identify the plasma metabolites with a different relative abundance in the metabolome at pre-dose and when the desvenlafaxine peak plasma concentration was reached (pre-dose vs. post-dose).
New Sentence: Next, ultra-performance liquid chromatography-quadrupole time-of-flight mass spectrometry was used to identify plasma metabolites with different relative abundances in the metabolome at pre-dose and when the desvenlafaxine peak plasma concentration was reached (pre-dose vs. post-dose).
Comments 3: In Table S1, the row of sex has mean 35 and standard deviation 51-49. Could you please give more clarifications around that?
Response 3: Thank you for noting. Of the total number of 35 individuals, 18 are female and 17 are male, with their proportion being 51% and 49%. This has been reviewed and corrected in Table S1.
Comments 4: From Line 303 to line 342, the author discussed about the association to PK. But no data in the article have indicated any correlation. I would appreciate more clarification.
Response 4: We appreciate the reviewer's comments. The first section of the Integrated pathways and community network analysis results (Table 2) presents a set of the differential metabolites associated with pharmacokinetic parameters that we selected because of their significance and evidence using pathway enrichment. The interactions can be observed in Figure 3 and the details in table S5 and S6. Using these results, we discussed the group of metabolites with some functional association with desvenlafaxine or general pharmacokinetics in the section pointed out by the reviewer. This could be used and validated for clinical significance in future studies to assess the pharmacology of inter-individual responses.

Reviewer 2 Report
Comments and Suggestions for Authors
This manuscript presents a comprehensive metabolomic analysis following a single venlafaxin administration in healthy volunteers, suggesting that already after a single dose a broad metabolomic response is observed, which may explain different inter-individual response and adverse effect profile. It is unclear how these results may relate to longer-term treatment, the antidepressant effect usually beeing observed with a delay of a couple of weeks. The analysis seems to be very comprehensive and innovative, however, not beeing familiar with metabolomics in detail, it remains unclear how this may improve clinical management. Importantly, figures are missing in the manuscript version submitted.
Comments on the Quality of English Languagegood
Author Response
Thank you very much for taking the time to review our manuscript. We appreciate all the suggestions you made. Please find our detailed responses below, with the corresponding revisions and corrections highlighted in red in the resubmitted manuscript. We would like to apologize for the problems with visualizing the figures in the manuscript. In fact, we forward them individually and separately to the text, and they end up not being viewed. We regret any confusion this may have caused.
Point-by-point
Comments 1: This manuscript presents a comprehensive metabolomic analysis following a single venlafaxin administration in healthy volunteers, suggesting that already after a single dose a broad metabolomic response is observed, which may explain different inter-individual response and adverse effect profile. It is unclear how these results may relate to longer-term treatment, the antidepressant effect usually beeing observed with a delay of a couple of weeks. The analysis seems to be very comprehensive and innovative, however, not beeing familiar with metabolomics in detail, it remains unclear how this may improve clinical management. Importantly, figures are missing in the manuscript version submitted.
Response 1: We appreciate the reviewer's comments. Our work brings a very comprehensive approach correlating metabolomic, pharmacokinetic and clinical data. As a first approach, we identified those metabolites whose difference in abundance between pre-dose and post-dose was statistically significant. Using this 'set of differential metabolites', we established apre-dose and post-dose associations with pharmacokinetic parameters and clinical/laboratory parameters post-study. In this sense, we evidenced a group of metabolites with some predictive or explanatory power of the effects of a unique dose of DVS on metabolism. Importantly, this could be used and validated for clinical significance in future studies to assess response pharmacology only at the pre-dose stage. However, we agree with the reviewer that our data is limited to discussing possible effects directly related to the mechanism of action of desvenlafaxine. Future studies are still necessary in this context.

Reviewer 3 Report
Comments and Suggestions for Authors
This paper explores the effects of a single dose of desvenlafaxine -via oral administration- on pharmacokinetic parameters, and clinical as well as laboratory characteristics in healthy volunteers using a pharmacometabolomic approach. Overall, the manuscript provides valuable information, and the methodology has potential therapeutic use. However, before being considered for publication in Pharmaceutics, a few important concerns must be addressed by authors.
1. Authors failed to represent any experiment results in the abstract part that would increase the scientific interest. Authors should include the PK profile or metabolomics outcomes in this section.
2. Authors should include more references on desvenlafaxine physicochemical properties and its related research on pharmacodynamic and pharmacometabolomic study in the introduction section.
3. I did not find any figures in the manuscript. Please check it.
4. Representative chromatogram for respective blank and analytes is missing. It should be supplemented in the revised manuscript
5. The authors have lots of sampling points and withdraw 8 mL of blood at each time point. So, could authors provide an explanation regarding the potential impact of rapid blood ejection on pharmacokinetic profiling?
6. Did the authors follow the USFDA guidelines for method validation? Please justify and provide the extraction recovery of analytes in the revised version.
Comments on the Quality of English Language
. The grammatical and typo errors should be verified throughout the manuscript.
Author Response
Thank you very much for taking the time to review our manuscript. We appreciate all the suggestions you made. Please find our detailed responses below, with the corresponding revisions and corrections highlighted in red in the resubmitted manuscript. We would like to apologize for the problems with visualizing the figures in the manuscript. In fact, we forward them individually and separately to the text, and they end up not being viewed. We regret any confusion this may have caused.
Point-by-point
Comments 1: Authors failed to represent any experiment results in the abstract part that would increase the scientific interest. Authors should include the PK profile or metabolomics outcomes in this section.
Response 1: We agree with the reviewer and a new sentence referring to this information has been included in the text.
New sentence: “Correlations were observed between metabolomic profiles, such as Tyrosine, Sphingosine 1-phosphate and pharmacokinetic parameters, as well as Acetoacetic acid, Uridine diphosphate glucose associated with clinical characteristics. Our findings suggest that desvenlafaxine may have a broader effect than previously thought by acting on proteins responsible for the transport of various molecules at the cellular level, such as the solute carrier SLC and adenosine triphosphate synthase binding cassette ABC transporters. Both of these molecules have been associated with PK parameters and adverse events in our study.”
Comments 2: Authors should include more references on desvenlafaxine physicochemical properties and its related research on pharmacodynamic and pharmacometabolomic study in the introduction section.
Response 2: We agreed with the reviewer, and new information was included in the introduction. To our understanding, no specific articles are commenting on the use of metabolomics in the study of desvenlafaxine pharmacokinetics. However, in agreement with the reviewer, we have added some information in the introduction. If the reviewer feels it’s necessary to add additional information to the introduction, please suggest further changes.
Old sentence: Desvenlafaxine belongs to the serotonin-norepinephrine reuptake inhibitor (SNRI) class and is marketed as Pristiq®.
New sentence: Of these, Desvenlafaxine belongs to the serotonin-norepinephrine reuptake inhibitor (SNRI) class and is marketed as Pristiq®, was approved by the FDA in 2008 to treat major depressive disorder (MDD) in adults. As a selective SNRI, desvenlafaxine does not significantly affect muscarinic-cholinergic, H1-histaminergic, or α1-adrenergic receptors, nor does it inhibit monoamine oxidase (2). This selective inhibition is thought to contribute to its therapeutic effects in managing depression (3).
Old sentence: Structurally, Desvenlafaxine succinate (DVS) is the succinate salt monohydrate of O-desmethylvenlafaxine, an active metabolite of venlafaxine (C16H26NO2) that can be excreted unchanged (~45%), as the glucuronide metabolite (~19%), and <5% as the oxidative metabolite (N,O-didesmethylvenlafaxine). It is oxidized primarily by CYP3A4 (3). Upon oral intake, its bioavailability stands at 80%, reaching maximum plasma concentrations (Cmax) after 7.5 hours, and it shows low plasma protein binding (30%). The distribution of the drug after intravenous administration suggests extensive non-vascular compartmentalisation, with a volume of distribution of 3.4 L/kg and a mean half-life of approximately 11 hours (3). Traditional pharmacokinetic prediction methods (4) are not always able to predict the behavior of the drug in a given individual. Metabolites, which are the chemical entities that are transformed during metabolism, provide a functional readout of the cellular biochemistry that is the best predictor of the phenotype (5). It's justifies the relevance of a pharmacometabolic approach to explore the pharmacokinetic variability of desvenlafaxine in healthy volunteers.
New sentence: Structurally, Desvenlafaxine succinate (DVS) is the succinate salt monohydrate of O-desmethylvenlafaxine, an active metabolite of venlafaxine (WM: 263.3752g/m, C16H25NO2) that can be excreted unchanged (~45%), as the glucuronide metabolite (~19%), and <5% as the oxidative metabolite (N,O-didesmethylvenlafaxine). It is oxidized primarily by CYP3A4 (5). Upon oral intake, its bioavailability stands at 80%, reaching maximum plasma concentrations (Cmax) after 7.5 hours, and it shows low plasma protein binding (30%). The distribution of the drug after intravenous administration suggests extensive non-vascular compartmentalisation, with a volume of distribution of 3.4 L/kg and a mean half-life of approximately 11 hours (5). Traditional pharmacokinetic prediction methods (6) are not always able to predict the behavior of the drug in a given individual and inter-individual variability in response to this drug remains a clinical challenge (7). Pharmacokinetic and pharmacometabolic studies are essential to improve our understanding of the mechanisms underlying this variability. The current literature, although comprehensive, could benefit from further exploration of metabolomic. Metabolites, which are the chemical entities that are transformed during metabolism, provide a functional readout of the cellular biochemistry that is the best predictor of the phenotype(5). It justifies the relevance of a pharmacometabolic approach to explore the pharmacokinetic variability of desvenlafaxine in healthy volunteers.
New Order References
References:
2. Lieberman DZ, Massey SH. Desvenlafaxine in major depressive disorder: an evidence-based review of its place in therapy. Core Evid. 2009;4:67–82.
3. Deecher DC, Beyer CE, Johnston G, Bray J, Shah S, Abou-Gharbia M, et al. Desvenlafaxine Succinate: A New Serotonin and Norepinephrine Reuptake Inhibitor. J Pharmacol Exp Ther. 2006 Aug 1;318(2):657–65.
4. Norman TR, Olver JS. Desvenlafaxine in the treatment of major depression: an updated overview. Expert Opinion on Pharmacotherapy. 2021 Jun 13;22(9):1087–97.
5. I AN, A JB, Parks V, S LR, B SM, Posener J, et al. Pharmacokinetics, Pharmacodynamics, and Safety of Desvenlafaxine, a Serotonin-Norepinephrine Reuptake Inhibitor. Journal of Bioequivalence & Bioavailability. 2013;5(1):022–30.
6. Beger RD, Schmidt MA, Kaddurah-Daouk R. Current Concepts in Pharmacometabolomics, Biomarker Discovery, and Precision Medicine. Metabolites. 2020 Apr;10(4):129.
7. Pirmohamed M. Pharmacogenomics: current status and future perspectives. Nat Rev Genet. 2023 Jun;24(6):350–62.
Comments 3: I did not find any figures in the manuscript. Please check it.
Response 3: We thank the reviewer for the observation. The file with the figures was added separately from the text, so there must have been an error. We’ll incorporate it into the text as well.
Comments 4: Representative chromatogram for respective blank and analytes is missing. It should be supplemented in the revised manuscript.
Response 4: We thank the reviewer for pointing this out. To verify the selectivity assay for desvenlafaxine, we monitored the specific m/z transitions for the analyte and internal standard (IS) in a pooled blank plasma sample. Figure S1 shows the correspondent chromatogram, which allows us to ensure that there are no interfering peaks from the endogenous components occurring in the same retention time in blank plasma samples with both the analyte and IS.
Figura S1. Chromatograms for the selectivity assay. Analysis of desvenlafaxine-spiked (a) and blank (c) plasma samples e, monitored at channel 264.3>58.2. Analysis of IS-spiked (b) and blank (d) plasma samples monitored at channel 270.3>181.2.
Comments 5: The authors have lots of sampling points and withdraw 8 mL of blood at each time point. So, could authors provide an explanation regarding the potential impact of rapid blood ejection on pharmacokinetic profiling?
Response 5: We appreciate the reviewer's comments. Given that this study followed the guides from ANVISA (The Brazilian Health Regulatory Agency), this agency requires the collection of additional backup samples to ensure the availability for reanalysis, if needed. Consequently, it is essential to collect an 8 mL blood sample, preventively. Our center of bioequivalence has more than 20 years of experience in this study's model with the same purpose. Proper training and knowledge of the factors that may induce hemolysis as well as implementation of standardized collection procedures was used to minimize hemolysis rates and improve the reliability of laboratory results.
Comments 6: Did the authors follow the USFDA guidelines for method validation? Please justify and provide the extraction recovery of analytes in the revised version.
Response 6: Thank you for commenting on this. Although this study did not follow the USFDA guidelines, we did rely on the Brazilian national guidelines from ANVISA (the Brazilian Health Regulatory Agency), in accordance with regulation number 742 of 10/08/2022 (Resolution DC/ANVISA Nº 742 of 10/08/2022). Although there are variations in the evaluation criteria, ANVISA is internationally recognized for its excellence and is accepted as a model in several countries. For ANVISA guidelines, the calculation of recovery is based on the area's ratio of the analyte and its Internal Standard, resulting in the relative recovery. On the other hand, FDA guidelines recommend accessing the analyte recovery individually, resulting in the absolute recovery. For this reason, herein the absolute recovery assay was not performed.
Comments 7: The grammatical and typo errors should be verified throughout the manuscript.
Response 7: We appreciate the reviewer's suggestion. The entire manuscript was carefully reviewed and edited.

Round 2
Reviewer 1 Report
Comments and Suggestions for Authors
Thanks to the authors for the revision. Overall it looks good for publication except for some minor suggestions as mentioned below:
1. in the introduction, I am quite confused about WM: 263.3752g/m for venlafaxine. Dose the author mean MW? If so, its MW should be 277.402 g/mol. If the authors means desvenlafaxine, please update WM to MW
2. Axis in Figure 1 is not readable.
Author Response
Thank you very much for taking the time to review our manuscript. We appreciate all the suggestions you made. Please find our detailed responses below, with the corresponding revisions and corrections highlighted in red in the resubmitted manuscript.
1) Thanks to the authors for the revision. Overall it looks good for publication except for some minor suggestions as mentioned below:
Response: We thank the reviewer for their positive feedback.
2) In the introduction, I am quite confused about WM: 263.3752g/m for venlafaxine. Dose the author mean MW? If so, its MW should be 277.402 g/mol. If the authors means desvenlafaxine, please update WM to MW.
Response: We thank the reviewer for his comment. It has been revised and corrected in the manuscript.
Old sentence: Structurally, Desvenlafaxine succinate (DVS) is the succinate salt monohydrate of O-desmethylvenlafaxine, an active metabolite of venlafaxine (WM: 263.3752g/m, C16H25NO2) that can be excreted unchanged (~45%), as the glucuronide metabolite (~19%), and <5% as the oxidative metabolite (N,O-didesmethylvenlafaxine).
New sentence: Structurally, Desvenlafaxine succinate (DVS), an active metabolite of venlafaxine, is the succinate salt monohydrate of O-desmethylvenlafaxine, (MW: 263.1885g/mol, C16H25NO2) that can be excreted unchanged (~45%), as the glucuronide metabolite (~19%), and <5% as the oxidative metabolite (N,O-didesmethylvenlafaxine).
3) Axis in Figure 1 is not readable.
Response: We appreciate the reviewer's comments, the figure was updated.
